# The Impact of Chief Sustainability Officers on Environmental Performance of Korean Listed Companies: The Mediating Role of Corporate Sustainability Practices

Nebedum Ekene Ebele [1], Seong Mi Bae [2] and Jong Dae Kim [2],*

[1]  Sustainability Management Graduate Program, Inha University, Incheon 22212, Republic of Korea; ekenenebedum@gmail.com
[2]  College of Business Administration, Inha University, Incheon 22212, Republic of Korea; smbae@inha.ac.kr
*   Correspondence: jdk@inha.ac.kr

**Abstract:** Chief sustainability officers and sustainability consultants have become increasingly common today as many organizations have become more aware of their impact on the environment and society at large. With the growing importance of integrating sustainability into business, many firms are appointing chief sustainability officers to manage and oversee the sustainability affairs of their firms at various levels. However, very little is known about the chief sustainability officer and the sustainability management team. This study investigates the potential importance of the chief sustainability officer (CSO) and the sustainability management team toward the firm's environmental performance. Using a sample of Korean-listed companies for the year 2017–2020, this study aims to investigate the relationship between appointment of the chief sustainability officer and a firm's environmental performance. It also explores the possible mediating role of corporate sustainability practices (CSP) in this relationship, utilizing Baron and Kenny's method to analyze the mediating role. First, the regression analysis was conducted to assess the impact of the presence and role of CSO on the firm's environmental performance. Subsequently, the firm's CSP was then introduced into the regression analysis as a mediator, to evaluate its influence on the relationship between the chief sustainability officer and the firm's environmental performance. We found that CSP completely mediated the relationship between CSO and environmental performance. This study contributes empirically to the growing literature on the relevance of sustainability management officials and their impacts on the firm's environmental performance.

**Keywords:** chief sustainability officer (CSO); environmental performance; corporate sustainability practices; mediator

## 1. Introduction

Recently, companies have increasingly embraced sustainability management practices due to pressure from shareholders and the pursuit of competitive advantage. With environmental issues on the rise, sustainability management has become a critical practice for all organizations. The need to integrate sustainability into the top management responsibilities has become crucial. Consequently, many organizations are establishing sustainability management teams or committees managed by the Chief Sustainability Officer (CSO). This position is also referred to by other titles such as Chief Responsibility Officer, Corporate Social and Environmental Officer, and Executive or Senior Vice-president of Sustainability [1].

Creating a sustainability management team or committee, typically managed by a CSO, elevates sustainability to a high priority for the company, deeply rooted in its strategic thinking [1,2]. Therefore, the inclusion of CSO in the top management teams portrays a distinct manifestation of the firm's strategic response to shareholders and societal demands for sustainability concerns [3,4].

For the purpose of this research, we will use the term CSO to refer to all sustainability executives, and corporate sustainability practices (CSP) as the mediator that facilitates the CSO's responsibilities in enhancing the firm's environmental performance. Many shareholders are beginning to express their concerns over the existing traditional executive team's ability to fully grasp and integrate sustainability practices into the firm-specific strategies [5,6]. While the CEO primarily focuses on providing overall strategy and identifying market growth opportunities, the role of CSOs involves improving working conditions in the supply chain, creating better safety procedures and generating profits from products that address environmental and social problems [7].

Given that the CSO position and its team are relatively new in some firm's governance hierarchy, limited research has been conducted on the impact of the CSO on the firm's performance, particularly its contribution toward environmental performance. The relevance and contributions of CSO are devoid of much research. Most available research has focused on other top management roles and characteristics, while some have only briefly described the CSO's emergence [8,9].

Considering the growing dynamics and evolving nature of the CSO position, the research findings of [9] further emphasize the need to delve more into other factors related to the specific nature and contributions of the CSO appointments. However, the empirical research on how CSO presence affects a firm's environmental performance is still scant. Addressing this gap is important because the core purpose of having a CSO in a firm is to manage the firm's sustainability practices which have become increasingly important nowadays for firms hoping to achieve and maintain its competitive advantages [10].

Prior literature on CSOs mostly focuses on their impacts on sustainability performance. Among them, reference [11] focused on CSO and social performance, and [12] investigated the impact of CSOs on environmental performance. However, such research produced a mixed result which does not provide enough evidence to prove that CSO positively affects sustainability performance. And one possible explanation for these mixed results is the marginalization of CSOs when sustainability knowledge and experience are acquired externally [13,14]. Additionally, related literature found little evidence of the impact of sustainability committees on performance outcomes [15–17]. For instance, findings by [17,18] found no significant relationship between sustainability teams and environmental performance. The mixed and inconsistent findings from the previous literature form the basis for further investigation to gain a better understanding of the relationship between the variables in the present study.

To address the relationship between corporate sustainability practices and environmental performance, it is very important to use an appropriate measurement for environmental performance. This is so because existing literature has been consistently applying the KLD (Kinder Lydenberg Domini) rating since 1991 as a measure of environmental performance. Some recent research has recognized that the KLD strength and weakness dimensions do not fully represent a firm's environmental performance [19].

To address this limitation, this study takes a novel approach by developing a specific measurement construct for environmental performance. This construct is designed to capture a more holistic view of a firm's environmental performance and can be analyzed using content analysis coding. The variables and measurement items developed for this purpose are detailed in the Appendix A.

Therefore, the uniqueness of this research model lies in its departure from the conventional KLD rating system, aiming to provide a more comprehensive and nuanced understanding of a firm's environmental performance through a tailored measurement approach. This new approach allows for a more accurate assessment of the relationship between corporate sustainability practices and environmental performance, potentially yielding valuable insights that previous research may have overlooked.

The primary goal of this study is to enhance our understanding of the connection between CSO and environmental performance. We aim to achieve this by investigating the role of corporate sustainability practices as a mediator in this relationship. This research is

particularly relevant given the increasing emphasis on integrating sustainability practices into a firm's business model [20].

With the growing importance and focus on sustainability management officer positions, our study on CSO and its impact on environmental performance contribute to the corporate governance literature by elaborating on the importance of CSOs and their roles in enhancing organizational sustainability performance. By examining the relationship between the CSO and environmental performance, this study will provide valuable insights that support the managerial attention given to different decision domains, as enabled by the specialized executives, which ultimately drives firm outcome in those domains [11,21].

The structure of the paper is organized as follows. Section 1 introduces our research and its objectives. Section 2 provides a comprehensive literature review of the main variables in this study, and also the theoretical background and hypothesis development. Section 3 outlines our research samples and methodology. The research results are discussed in Section 4, while Section 5 covers the research theoretical and practical implications. Finally, Section 6 offers a conclusion based on the research outcomes.

**2. Literature Review**

This section sheds light on our key variables which are chief sustainability officer, environmental performance and corporate sustainability practices. This section will further explain the theoretical support, hypothesis development and conceptual framework for our study.

*2.1. Chief Sustainability Officer & Sustainability Management Team*

To gain a comprehensive understanding of CSOs, it is important to consider various definitions from the existing literature. Typically, CSOs are senior executives in top management teams who are specifically responsible for managing a firm's corporate social performance [22].

According to [1], a CSO is defined as an executive who is responsible for taking the primary responsibility of the corporate sustainability or issues related to the corporate social performance of the firm [9]. It is worth noting that the CSO's responsibilities may also vary across industries, firms and the stages of firm development, but, despite these differences, most of them share core responsibilities in corporate social performance [23].

In general, the core roles of the CSOs are to formulate, oversee and execute the firm's sustainability strategy and activities. They are responsible for reviewing business practices, analyzing social impacts and formulating proper strategies that integrate profit growth and sustainable development of the firm. Additionally, CSOs are often responsible for managing stakeholder relationships and fostering a culture of sustainability within the firm. The CSO is also a part of the top management executive of the firm, thus can be found at the top of the corporate hierarchy at the C-suite level and the CSO reports directly to the CEO or the board of directors [8]. The CSO position and team are mostly found in firms or industries that are regarded as environmentally sensitive.

To further elucidate the importance of a CSO, Peters [24] stated that the establishment of a CSO position in a firm implies responsibility for performance outcomes and the presence of these executive positions reflects internal power shifts and accountability. The CSO presence will likely raise the level of sustainability responsiveness on the executive suite agenda and, in turn, change the firm's business practice.

As part of the role of the CSO position, the officer is charged with the task of managing and overseeing the sustainability management team or committee within the firm. To facilitate the decision-making process and execution of activities, the board often establishes a dedicated committee to focus on sustainability issues [25]. The existence of sustainability management teams or committee in various firms creates an environment that promotes greater legitimacy for sustainability concerns. This encourages key decision-makers and all members of top management to actively address the social issues faced by the organization [26].

## 2.2. Corporate Sustainability Practices

Corporate Sustainability practices (CSP) are increasingly considered to be important because of their contributions to firms in the context of gaining competitive advantage. CSP is recognized as an impressive and vital practice for firms all over the world to gain competitive advantage in this resource constrained twenty-first century [27].

According to [28], CSP is said to be an updated concept of social responsibility (CSR). Corporate sustainability is regarded as one of the most used concepts to describe the role of business in society [23]. When firms adopt and practice corporate sustainability, the latter is expected to contribute to the wellbeing of societies and reduce the adverse impacts of the firm's operations on societies [29]. When corporate sustainability is sustainably practiced, the firms enjoy some benefits from this process which range from reduced cost and risks to gaining a competitive advantage, developing a reputation and legitimacy, and win-win opportunities through creating values [30].

Many definitions have been proposed for sustainability, but the most widely accepted is the one given by the Bruntland report. The report led by Bruntland, the former prime minister of Norway, defines sustainable development "as the development that meets the needs of the current generation without compromising the needs of the future generations to meet their own needs" [31].

The term corporate sustainability practice is seen as a complex phenomenon that comprises of three aspects of business activities such as economic, social and environmental [32]. It embodies the idea that human society should operate by applying industrial and biological processes that can be sustained indefinitely [33]. It proposes that for a firm to be sustainable it needs to adopt good practices by behaving in a socially and environmentally responsible manner without neglecting its financial objectives [34].

For this study, we will define corporate sustainability practices as a business concept and strategy that incorporates the core objectives of the triple bottom line (TBL)—economic, environmental and social contribution into its corporate business model to achieve long-term competitive advantage and performance.

## 2.3. Environmental Performance

Environmental performance is regarded as a special form of corporate social responsibility. It measures how effective a firm is in its actions and strategies in minimizing its impact on the environment [35]. Environmental performance or strategy can be referred to as "a firm's strategy to manage the interface between its business and the natural environment" [36]. Environmental performance is a reflection of a firm's environmentally related processes, policies, programs and observable outcomes [37]. Corporate environmental activities can vary substantially from beyond-compliance proactive approaches that require firms to build specific capabilities and resources to reactive solutions that minimally meet (or fail to meet) regulatory standards [38].

Environmental performance issues have become pertinent to companies because of the values such practices add to companies by saving costs [39] and reducing systematic risk through conformance with institutional expectations that give firms access to resources and protect them from scrutiny [40]. The integration and implementation of environmental performance strategies by firms into their core business strategy allows the firms to develop valuable, inimitable and non-substitutable organizational resources and capabilities [41], enabling firms to create competitive advantage.

## 2.4. Related Theories and Hypothesis Development

This section explains the theoretical foundation and support for the direct relationship between CSO and firm environmental performance. The Upper Echelons theory was utilized to support the intertwined relationship between sustainability management officers and environmental performance, as well as the introduction of CSP. We also drew some reference from Legitimacy theory to support the firm's environmental performance.

### 2.4.1. Upper Echelons Theory

The Upper Echelons theory, as originally proposed by [42], states that an organization's strategic choices and performance levels are partially predicted by managerial background and characteristics. The Upper Echelons main idea is that the board managers choose firm strategies and through these strategies they influence the firm's outcome [43].

This theory is best suited to the investigation of the position and role of the C-level officers or top board management officers with specific function domains. From the Upper Echelons theory view, the CEO turnover can transform the corporate strategic orientation of a firm [44].

These top management executives have been shown to contribute to strategic change [45], corporate social strategy [13,46] and firm innovation and differentiation [47]. The Upper Echelons theory provides a framework to study the existing antecedents of top management attitudes and behaviors regarding corporate sustainability, especially the appointment of chief sustainability officers to the top management team [10,48].

The Upper Echelons theory mostly emphasizes the pertinent role of the board management in impacting organizational outcomes through its strategies. This theory maintains that the presence and role of the CSO and its team facilitate the design and implementation of environmental strategy because the CSO has the capability and potential required to oversee environmental issues needed to set a long-term environmental development plan for the firm. Hence, with the inclusion of the CSO in the top management and also the role of the CSO in the organization's board, this study adopts the Upper Echelons theory as one of its main theoretical frameworks

### 2.4.2. Legitimacy Theory

Legitimacy theory is another theory that supports this study in relation to environmental performance. This theory asserts that firms must try to conform to the norms of their operating society and the environment in which they run their business [49]. Environmental concerns are important issues that constitute part of a firm's public legitimacy and reputation in its environment [50]. Firms adhering to their operating environment norms can be regarded as conforming to the responsibility to the public and society. Furthermore, this theory argues that firms can actively seek and maintain their legitimacy by aligning company values, policies and strategies with community values. By doing so, they use their environmental performance as a strategy to gain legitimacy from stakeholders [51].

For an organization to gain its legitimacy, the establishment of a CSR committee and/or the appointment of a CSO position within the company is essential, as this board's expertise may encourage a company to prepare a sustainability or CSR report voluntarily and, in doing so, improve its CSR-related reputation and disclosure [52]. Based on the Legitimacy theory, through social contracts and regulations, a company adheres to society's values and expectations [53]. As societal norms and regulations change, companies have to establish strategies through which they gain legitimacy for the CSR orientation to corporate activities [54].

### 2.5. Hypothesis Development

In this section, we explain the hypothesis underlying the relationship among our key variables: chief sustainability officer, organizational performance and corporate sustainability practices.

### 2.5.1. CSO and Environmental Performance

The CSO is a position that has the capacity to guide the direction of a company toward sustainability practices, which ultimately should lead to financial as well as environmental performance improvements [2]. This position is believed to have a substantial influence on the effective implementation of sustainability strategies and initiatives. The creation and inclusion of the CSO in the top management team indicates to the stakeholders that the firm

is making an appropriate commitment to improving the effectiveness of the management in the area of sustainability management.

Previous literature has argued that the presence of a CSO is likely to enhance organizations' internal and external practices and processes that will further improve the environmental performance of the firms [55].

Reference [56] investigated and approved the effects of board-level sustainability committees on sustainability performance. Also, references [26,57] provided evidence that the creation of specialized sustainability committees leads to superior sustainability performance.

Another study by Peter [25] examined the relationship between CSO expertise and sustainability performance and documented a mixed association between sustainability performance outcomes after the appointment of CSOs. [58] Stated that appointing a chief of CSR is an important milestone to move companies through the corporate responsibility stage. Hence, it will improve a company's reputation and, in turn, might affect its performance positively. CSOs as the coordinators of firm's sustainability strategies, with their wide range of environmental expertise, can contribute to the decision-making team, a phenomenon which leads to setting a long-term environmental plan for the company. In addition to existing similar literature, this study will empirically investigate the relationship between the presence of CSO and environmental performance. Thus, we propose the hypothesis as below:

**H1.** *The CSO's presence and action will positively influence the environmental performance of the firm.*

### 2.5.2. CSO and Corporate Sustainability Practices

The presence of a CSO and sustainability management team has been found to be central to a firm's sustainability practices and decision-making for sustainable strategic visions and missions [10]. The role of the CSO and sustainability management team or committee has become very important and central through the need for substantial organizational changes necessitated by the adoption of corporate sustainability practices [59]. According to a prior study, a firm that has set up a team or committee especially for sustainability issues and practices exposes its senior executive to more frequent discussion about the importance and urgency of improving the firm's social sustainability performance [60]. Previous empirical studies found a positive relationship between sustainability committees managed by sustainability officers and CSP [61]. Also, a firm's effectiveness toward sustainability practices can be significantly improved with the advice and support of the specialized committee, thereby contributing to advanced social performance [62].

In the growing context of corporate sustainability practices, the CSO and sustainability management team play an important decisive role. Through its set of formal and informal rules and procedures, the CSO and its team shape the firm's sustainability practices. The CSO as a specialized top executive can serve as a key player for all issues related to corporate sustainability practices. Hence, the enhancement of corporate sustainability practices within a firm is contingent upon the effectiveness of its CSO and team. Consequently, we put forth the following hypothesis:

**H2.** *The presence of a CSO and its team will significantly enhance the firm's corporate sustainability practices.*

### 2.5.3. Corporate Sustainability Practices and Environmental Performance

Corporate sustainability practices, with their growing adoption by many organizations, have been regarded as the role of businesses in society [24]. The idea of corporate sustainability practices requires firms to incorporate and interconnect environmental, social and economic issues at different levels [63].

Generally, firms practicing corporate sustainability practices are expected to contribute to the well-being of societies and reduce the adverse impacts of their operations on them [30]. For the benefits of engaging in corporate sustainability, the involved firms enjoy certain

dividends ranging from reduced risk and costs to gaining competitive advantage and other win-win outcomes through creating value [31]. The adoption and implementation of corporate sustainability practices require firms to take their environmental and social impacts into account in alignment with their economic objectives [64].

Previous literature has studied the relationship between sustainability practices and the environmental performance of the firms engaging in sustainability practices. Reference [65] suggests that effective application of corporate sustainability practices and strategies will improve organizational outcomes. Also, references [66,67] showed that sustainability-related strategic practices will lead to greater sustainability performance. A recent study from [68] with UK firms showed that firms with effective corporate sustainability practices or corporate strategies showed better sustainability performance.

Considering the fact that firms adopting corporate sustainability practices anticipate substantial improvements in their environmental performance, this study further investigates the relationship between corporate sustainability practices and environmental performance. Thus, we propose the following hypothesis:

**H3.** *Corporate sustainability practices will positively influence the firm's environmental performance.*

### 2.5.4. CSO and Environmental Performance: The Mediating Effect of CSP

A CSO hold a crucial role in influencing the strategic direction of a firm, potentially leading to a competitive advantage through their engagement in corporate sustainability practices. However, findings by [55] show a negative association between the presence of a CSO and environmental performance, indicating that, in some instances, CSOs may be hired for reasons unrelated to genuine sustainability purposes.

To elucidate how CSOs can positively impact their firm's environmental performance, we introduce the mediating variable of CSP (Corporate Sustainability Practices). We posit that effective implementation of CSP can enhance a firm's environmental performance when CSOs prioritize comprehensive CSP programs and initiatives. While prior literature has seldom empirically examined this relationship, recent studies such as [69,70] demonstrated the mediating role of CSP in different contexts, highlighting its potential positive influence on overall firm performance.

Building on this limited literature, our study seeks to explore the rarely investigated mediation of CSP in the context of the relationship between CSO presence and a firm's environmental performance. The introduction of CSP as a mediator will shed light on whether it indeed mediates the connection between CSO presence and environmental performance. If a significant indirect effect is observed, it would suggest that corporate sustainability practices play a pivotal role in translating the presence of a CSO into tangible improvements in environmental performance.

CSP encompasses the three dimensions of corporate sustainability, namely economic, social and environmental practices, all of which have been acknowledged for their substantial contributions to a firm's overall performance. In alignment with this perspective, our study proposes the following hypothesis:

**H4.** *Corporate sustainability practices will mediate the relationship between CSO presence and a firm's environmental performance. A conceptual framework for this study was developed, as shown in the Figure 1 below.*

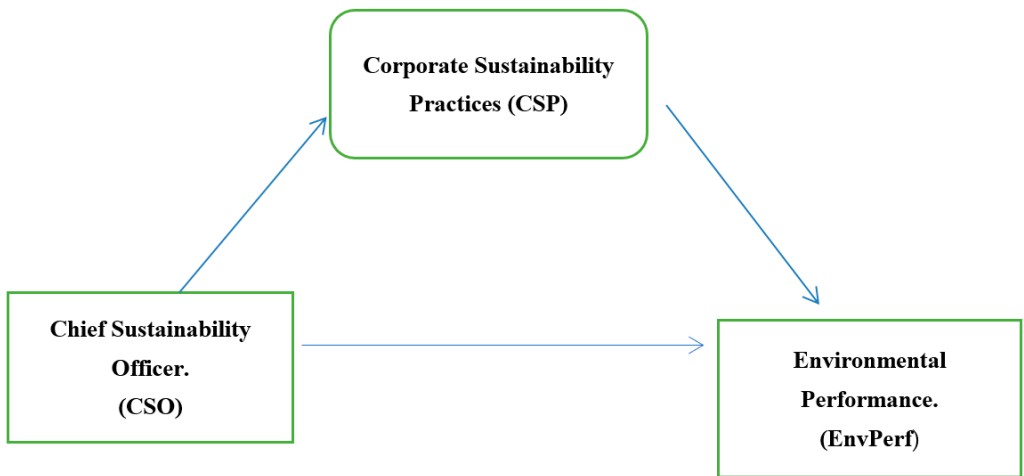

**Figure 1.** Conceptual framework for the study.

## 3. Samples and Methodology

### 3.1. Samples

This research investigated the impact of Chief Sustainability Officers (CSOs) on the environmental performance of firms, incorporating the mediating influence of Corporate Sustainability Practices (CSP). Our study consists of companies listed on the Korean stock exchange that voluntarily release sustainability reports. These reports serve as rich sources of data concerning environmental performance, the presence of CSOs, and CSP.

The sample predominantly comprises of major Korean top MNE corporations including industries such as construction, energy, automotive, telecommunications, technology and hardware, among others. From the pool of publicly listed Korean firms, we applied specific selection criteria, resulting in a final sample of 71 top firms, yielding a total of 284 observations spanning the years 2017 to 2020. This selection process ensured diverse observations, enhancing the statistical robustness of our analysis.

To quantify the variables in this study, we employed a quantitative approach and utilized content analysis as our research method. Content analysis was chosen for its flexibility in both data collection and analysis.

### 3.2. Measurement of Variables

3.2.1. Dependent Variable

For this study, firm environmental performance was investigated as the dependent variable. To measure environmental performance, content analysis was used instead of the common KLD (Kinder, Lydenberg and Domini) database, which only provides data from developed countries.

This study established a unique framework to evaluate a firm's environmental performance by developing ten items which were used as the coding criteria under content analysis. These ten coding items or constructs for environmental performance measure in this research were crafted by considering both internal and external aspects of a firm's environmental engagement. This included evaluating internal factors such as environmental compliance activities and external factors such as efforts to prevent environmental issues including emission reduction targets/plans, the firm's commitment to zero emissions, engagement in research and development, support for environmental organizations' activities, and the disclosure of environmental practices and sustainability reporting, which can be found in Appendix A.

Company sustainability reports were manually searched to record activities aligned with the environmental performance constructs developed.

For companies whose activities matched our environmental performance items, a score of "1" was given, while "0" was given when the activity was unrelated or unavailable. The total score of each construct was then used for the analysis.

### 3.2.2. Independent Variables

The CSO is the independent variable for this study. To determine the presence and effect of the CSO, the sustainability reports of the listed companies for fiscal years 2017–2020 were analyzed manually using content analysis. For the content analysis coding, six items were developed to measure the value of CSO. The content analysis coding construct for the CSO were developed based on the information regarding the CSOs as detailed in various firms' annual reports, as well as the roles and contributions of CSO team members. These details served as the foundation for constructing the coding framework for the CSO variable in this study. The constructs are listed in the Appendix A.

The value of "1" was assigned in case of presence of CSO/sustainability team and their specified duties as suggested by this study, and a value of "0" was assigned when no information was available. This coding method is consistent with assessments of sustainability officers from previous literature [25].

### 3.2.3. Mediating Variable

In this research, Corporate Sustainability Performance (CSP) was explored as the mediating variable. To access CSP, this study acknowledges the three pillars of sustainability practices: economic, social and environmental, which are indicative of a company's sustainable endeavors. Moreover, this study developed a set of 11 items to measure CSP; these items combined the economic, social and environmental initiatives undertaken by the companies. These 11 items were employed in the content analysis coding for CSP using information from the companies' sustainability reports.

### 3.2.4. Control Variables

In analyzing the relationship between the appointment of CSO and environmental performance, it is important to take into consideration the key control variables that may affect the result of the analysis. These selected control variables consist of financial and company characteristics. They were selected because they serve the purpose of mitigating potential biases in the results. These control variables were also chosen based on previous research [19,71,72].

For this study, various control variables such as firm size, leverage, board size, and market-to-book ratio were adopted. The market-to-book ratio was used as a proxy for firm growth. We also controlled for industry and year dummy variables.

The *firm size* was measured as the natural logarithm of total assets. Firm size was selected because it is related to the economic scale or scope, which may be important for competitive aspects [12].

*Return on assets* (ROA) was measured as the ratio of net income to total assets, and *leverage* was included to help control for financial stability of the firms [73] and it was measured as the total debts divided by the total assets of a firm. *Market-to-book ratio,* which is a proxy for firm growth, was measured as the total market capitalization divided by total equity. Finally, the *board size* variable consisted of the total number of directors in the board of directors of the firm, including both the inside and outside directors.

### 3.3. Model Specification

To investigate the mediating impact of CSP on the relationship between CSO and a firm's environmental performance, regression analysis based on [74] was conducted. For the statistical analysis, regression analysis was utilized to test the hypotheses in this research, and the SPSS statistical package was used to run the data analysis. Regression analysis is one of the most used statistical methods in research for the testing of the relationship between a dependent variable and one or more independent variables. The following analytical models are specified with the variable codes.

To test Hypothesis 1:

$$EnvPerf = \beta0 + \beta1CSO + \beta2FirmSize + \beta3Leverage + \beta4ROA + \\ \beta5M2BOOK + \beta6YearDummy + \beta7IndustryDummy + \varepsilon \tag{1}$$

To test Hypothesis 2:

$$CSP = \beta0 + \beta1CSO + \beta2FirmSize = \beta3Leverage + \beta4ROA + \beta5M2BOOK + \\ \beta6YearDummy + \beta7IndustryDummy + \varepsilon \tag{2}$$

To test Hypothesis 3:

$$EnvPerf = \beta0 + \beta1CSP + \beta2FirmSize + \beta3Leverage + \beta4ROA + \beta5M2BOOK + \\ \beta6YearDummy + \beta7IndustryDummy + \varepsilon \tag{3}$$

To test Hypothesis 4:

$$EnvPerf = \beta0 + \beta1CSO + \beta2CSP + \beta3FirmSize + \beta4Leverage + \beta5ROA + \\ \beta6M2BOOK + \beta7YearDummy + \beta8IndustryDummy + \varepsilon \tag{4}$$

## 4. Results

*Descriptive Analysis*

Table 1 presents the descriptive statistics and overview of the range and distribution of variables used in this study. The environmental performance had a minimum score of 1 and a maximum score of 10, with a mean score of 7.01, indicating that the companies in the sample had relatively high levels of environmental performance. The CSO variable had a mean value of 2.59, indicating that, on average, each company in the sample had about two and half sustainability-related officers or team members. The CSP variable had a maximum value of 11 and a mean value of 8.20, suggesting that most of the sample firms had high levels of corporate sustainability practices.

**Table 1.** Descriptive statistics.

| Variables | N (Obs.) | Mean | Minimum | Maximum | STD. Deviation |
|---|---|---|---|---|---|
| ENVPERF | 284 | 7.01 | 1 | 10 | 1.616 |
| CSO | 284 | 2.59 | 0 | 7 | 2.588 |
| CSP | 284 | 8.20 | 0 | 11 | 3.359 |
| ROA | 284 | 2.95 | −17 | 53 | 7.190 |
| FIRMSIZE | 284 | 7.74 | 5 | 11 | 0.817 |
| LEVERAGE | 284 | 1.78 | 0 | 50 | 4.451 |
| M2BOOK | 284 | 0.00 | 0 | 0 | 0.001 |
| BOARDSIZE | 284 | 7.90 | 0 | 15 | 2.285 |

Abbreviations: ENVPERF—environmental performance; CSO—chief sustainability officer; CSP—corporate sustainability practices; ROA—return on assets; FIRMSIZE-firm's size by; LEVERAGE- firm's leverage; BOARDSIZE-firm's board size; M2BOOK—market to book ratio.

Regarding the control variables, the firm size varies from a minimum value of 5 to a maximum value of 11, with a mean value of 7.744, indicating that most companies in the sample were of medium-to-large size. The ROA varies from a minimum value of −17 to a maximum value of 53, with an average value of 2.95. The minimum, maximum and mean values of leverage are 0, 50 and 1.78, respectively, indicating that the companies in the sample have relatively low levels of leverage. The minimum, maximum and mean value for market-to-book ratio is 0, 0.1 and 0.0012, respectively. Finally, board size had a maximum value of 15, a mean value of 7.90 and a standard deviation of 2.28, indicating that companies in the sample had a wide range of board sizes.

Table 2 presents the pairwise correlations among all variables within the models. The correlation coefficients reveal statistically significant positive associations between CSO

and CSP with environmental performance at ($p < 0.001$), providing support for H1 and H3. This implies that the presence and involvement of a CSO within a firm contribute to its environmental performance, and the adoption of sustainability practices exerts a positive influence on the firm's environmental performance. These findings align with prior research, underscoring the significance of effective corporate sustainability practices in enhancing a firm's sustainability performance [3,36].

**Table 2.** Correlation matrix.

| | ENVPERF | CSO | CSP | ROA | Firm Size | Leverage | M2B | Board Size |
|---|---|---|---|---|---|---|---|---|
| EnvPerf | 1 | | | | | | | |
| CSO | 0.235 *** | 1 | | | | | | |
| CSP | 0.234 *** | 0.124 * | 1 | | | | | |
| ROA | 0.124 * | 0.090 | 0.005 | 1 | | | | |
| Firm Size | −0.002 | 0.187 ** | −0.152 * | −0.284 *** | 1 | | | |
| Leverage | −0.046 | 0.048 | −0.131 * | 0.305 *** | −0.138 * | 1 | | |
| M2B | −0.038 | 0.051 | −0.115 | 0.158 ** | −0.104 | −0.068 | 1 | |
| Board Size | 0.162 ** | 0.143 * | 0.116 | −0.046 | 0.184 ** | −0.056 | 0.055 | 1 |

Correlation is significant at 0.1% (***), 1% (**) and 5% (*). EnvPerf—dependent variable; CSO—independent variable; CSP—mediating variable; and ROA, Firm Size, Leverage, M2B, and Board size are the control variables.

Additionally, CSO demonstrates a positive correlation with CSP at a significance level of 5%. Furthermore, all Variance Inflation Factor (VIF) values are below 3, indicating the absence of substantial multicollinearity in the model [75].

Table 3 presents the regression analysis conducted to examine the relationship among key variables, including the Chief Sustainability Officer (CSO), Corporate Sustainability Practices (CSP), and firm Environmental Performance (EnvPerf). The analysis involved several regression models to test our proposed hypotheses.

Model 1 examined the main effect of CSO presence on the firm's environmental performance. The result provides empirical support for H1, indicating a positive relationship between the presence of CSO and environmental performance. The p-value associated with this relationship is 0.085, suggesting that it is statistically significant. This finding aligns with previous research by [57,76], which also found that the presence of sustainability leadership, such as a sustainability committee, is associated with superior environmental performance.

Model 2 was designed to test H2, which posits a positive association between CSO presence and Corporate Sustainability Practices (CSP). The empirical results support H2, revealing a statistically significant positive relationship between CSO presence and CSP. The p-value associated with this relationship is 0.002, providing strong evidence that the presence of a CSO is associated with enhanced corporate sustainability practices. This finding underscores the role of the CSO in driving sustainability initiatives within the organization.

In Model 3, we assessed H3, which posits a positive relationship between CSP and EnvPerf. The empirical result further supports H3, indicating a statistically significant positive relationship between CSP and EnvPerf. This outcome strengthens the notion that effective corporate sustainability practices are indeed linked to improved environmental performance. This alignment between sustainability practices and environmental performance proves the importance of sustainability initiatives in enhancing a firm's environmental outcomes.

Since all three hypotheses are supported by the empirical results, we proceeded to test the mediating relationship hypothesis based on [74].

Reference [74] state that for a variable to be fully regarded as a mediator it will have to meet the three important conditions for mediation. First, the independent variable (CSO) should significantly affect the dependent variable (EnvPerf). Second, the independent variable should significantly affect the mediator variable (CSP). Lastly, when both the independent and mediator variables are included in a model, the value of the mediator variable should be significant, but the coefficient of the independent variable should be significantly reduced (partial mediation) or become insignificant (complete mediation).

**Table 3.** Result of regression analysis.

| Variables | Model 1 | Model 2 | Model 3 | Model 4 |
|---|---|---|---|---|
| Constant | 5.979 *** (6.227) | 13.909 *** (6.608) | 4.681 *** (4.649) | 4.881 *** (4.783) |
| CSO | 0.069 * (1.726) | 0.278 ** (3.183) | | 0.047 (1.169) |
| CSP | | | 0.085 ** (3.170) | 0.079 ** (2.893) |
| Control Variables | | | | |
| ROA | 0.022 (1.601) | 0.011 (0.359) | 0.021 (1.590) | 0.021 (1.559) |
| Firm Size | −0.128 (−1.076) | −0.942 (−3.625) | −0.032 (−0.268) | −0.053 (−0.444) |
| Leverage | −0.030 (−1.417) | −0.134 ** (−2.933) | −0.017 (−0.804) | −0.019 (−0.907) |
| Market-2-Book | −59.471 (−0.813) | −437.457 ** (−2.727) | −17.493 (−0.240) | −24.939 (−0.341) |
| Board Size | 0.165 *** (4.003) | 0.245 ** (2.711) | 0.150 *** (3.650) | 0.146 *** (3.534) |
| Industry Dummy | Yes | Yes | Yes | Yes |
| Year Dummy | Yes | Yes | Yes | Yes |
| Adjusted R | 0.193 | 0.102 | 0.213 | 0.214 |
| Observation | 284 | 284 | 284 | 284 |

Significance levels: *** $p < 0.01$, ** $p < 0.05$, * $p < 0.1$. Note: this table presents the empirical results obtained from the regression analysis. The first column contains a list of all the variables. The Model 1 column presents the result of the effect of CSO on EnvPerf. The Model 2 column presents the result of testing the effect of CSO on CSP. The model 3 column presents the result of regressing EnvPerf on CSP. The last column, model 4, presents the result of the mediation effect of CSP on the relationship between CSO and EnvPerf.

In Model 4, our study aimed to test Hypothesis 4 (H4), which proposes that Corporate Sustainability Practices (CSP) mediate the relationship between Chief Sustainability Officer (CSO) presence and firm Environmental Performance (EnvPerf)

The result of model 4 regression testing H4 showed that CSP had a complete mediating effect on the relationship between CSO and EnvPerf. Thus, this study confirms that the presence of a chief sustainability officer has an indirect, positive influence on environmental performance through the mediation of corporate sustainability practices.

To further confirm the mediation effect in our model, we conducted a Sobel test. The Sobel test result produced a Z-value of 2.872 with a *p*-value of 0.004.

The significant p-value result indicates that the mediation effect is statistically robust. The Sobel test provides additional empirical support for the notion that CSP serves as a mediator in the relationship between CSO presence and EnvPerf.

Regarding the control variables, the board size consistently demonstrated positive and significant coefficients across all models, implying that the boards of the firms in our sample comprised of numerous directors with a broader expertise and knowledge of sustainability practices and management whose contribution influences the firm's environmental performance.

## 5. Theoretical and Empirical Implications

Our study adopted the Upper Echelons theory and Legitimacy theory to support the relationship between CSO and environmental performance with CSP as a mediator. These theories have contributed some important theoretical implications to the relevant literature.

First, the theoretical contribution of Upper Echelons theory to this research lies in its ability to provide a conceptual framework to understand how the presence and role of a CSO within the top management team can influence a firm's environmental performance. In the context of this research, the Upper Echelons theory offers theoretical support for the idea that the appointment of CSOs to the top management team can lead to changes in a firm's strategic orientation, particularly in the domain of corporate sustainability. It suggests that CSOs, as part of top management, can impact the firm's strategic decisions and influence its environmental performance.

The theoretical contribution of Legitimacy theory to this study centers on its ability to provide a framework to understand how organizations operate through the establishment of CSR management and the appointment of CSOs, who actively seek and maintain legitimacy in the context of environmental performance. From the lens of this theory, it emphasizes that firms that align their values, policies and strategies with the community values are more likely to be seen as responsible and legitimate in the eyes of stakeholders.

In the context of this study, Legitimacy theory supports the idea that organizations can enhance their legitimacy by demonstrating a commitment to environmental performance. By establishing CSR management and appointing CSOs, the firms signal their dedication to sustainability.

Therefore, Legitimacy theory contributes to this study by providing a theoretical foundation to understand how firms manage and cope with changing societal expectations. Finally, this theory highlights the importance of aligning corporate strategies with societal values to gain and maintain legitimacy in the eyes of stakeholders.

Based on our study results, several practical implications can be derived.

First, this study contributes to the existing body of literature on the role of chief sustainability officers making important managerial contributions and recommendations for the improvement of environmental performance. That is, this study indicates that the inclusion of a CSO and sustainability team in the business decision may signal to the investors or stakeholders the firm's potential commitment to improving its environmental performance.

Second, this work contributes to policy implications for firms to include sustainability officers and teams in their governance structure and ensure the implementation of corporate sustainability practices to enhance environmental performance of the firms in South Korea.

Finally, this study proves that the presence of a CSO and its sustainability team reflects the firm's strong commitment to pursue sustainability performance.

## 6. Conclusions

This study scrutinizes and assesses the correlation between chief sustainability officers and the environmental performance of Korean firms. Furthermore, it explores the mediating influence of corporate sustainability practices on the connection between these two variables. The motivation behind this research stems from the growing significance of Chief Sustainability Officers (CSO). Consequently, the main objective of this study was to investigate the influence of the CSO on environmental performance.

First, the empirical analysis of the Korean companies that released their sustainability reports voluntarily revealed a positive impact of the presence of the Chief Sustainability Officer (CSO) on the firm's environmental performance. Our study aligns with significance of sustainability committees and officers in driving superior environmental performance [57,76].

Second, the study found a positive association between CSO and CSP. This finding strengthens the argument that CSOs are very important in facilitating a firm's commitment to sustainability practices. CSOs can be seen as an important figure in promoting and integrating sustainability into the firm's culture and system, and also in aligning the firm values with societal and stakeholder expectations [52]. Our work is in line with Legitimacy theory which posits that legitimacy toward society can be reached through various corporate governance mechanisms such as the presence of a CSO.

Most importantly, this study added a novel contribution to the literature by examining the mediating role of CSP in the relationship between CSOs and firm environmental performance. Based on our mediation analysis results, this study confirms that CSPs fully mediate the impact of CSOs on environmental performance. This result proves that, while CSOs are instrumental, their influence on environmental performance is channeled through the strategic implementation of sustainability practices. The study further shows that firms with effective CSP enjoy a positive impact on their environmental performance.

Furthermore, from our control variables, the board size showed a positive influence on environmental performance. This implies that the role of the board members is very important and largely relevant in addressing the firm's environmental concerns. Also, their presence and

contributions can improve the effective implementation of corporate sustainability practices through the appointment of chief sustainability officers and sustainability teams.

Although this study makes an important and definite empirical contribution to existing literature on sustainability management officers, there are still some limitations that need to be taken into consideration for further research. Firstly, given that our study was focused on chief sustainability officers and environmental performance, a set of theoretically guided constructs was created to measure these variables. It would be interesting to see further studies develop more measurement items or methods for these variables and explore additional potential variables.

Secondly, it is important to note that this study had a limited sample size consisting of selected Korean firms between 2017 and 2020, and the time span of the study was relatively short. Therefore, future research should aim to include a larger and more diverse sample of companies with a longer observation period. This would allow for a more comprehensive understanding of the long-term impact of CSOs on firm environmental performance. Expanding the scope of research in this area would be valuable in further exploring the relationships and variables studied. Given the significance and relevance of our research findings, we strongly recommend extending this line of inquiry in future studies.

**Author Contributions:** Conceptualization, N.E.E.; Formal analysis, N.E.E.; Investigation, J.D.K.; Resources, S.M.B.; Writing—original draft, N.E.E.; Writing review & editing, S.M.B.; Supervision, J.D.K.; Project administration, S.M.B.; Funding acquisition, J.D.K. All authors have read and agreed to the published version of the manuscript.

**Funding:** This research was funded by the Inha University.

**Data Availability Statement:** Not applicable.

**Conflicts of Interest:** The authors declare no conflict of interest.

## Appendix A. The Constructs or Items Used for Measuring the Variables for This Research

These constructs or items were used for the content analysis and a score of "1" was given when there is information related to any construct, and "0" when there is no related or available information.

| Variables | Construct/Items |
|---|---|
| Environmental Performance (EnvPerf) | ➢ Company set GHG emission reduction target plans for facilities.<br>➢ Commitment to long term energy efficiency (Zero emission/clean production).<br>➢ Development of R/D investment plan for production and innovation.<br>➢ Investment in green business strategy<br>➢ Initiating a reduction and investment goal for low-carbon.<br>➢ Supporting international climate change organization development projects.<br>➢ Involvement in sustainability reporting and disclosure practices (GRI&CDP) |
| Chief Sustainability Officer & Team | ➢ Presence of Sustainability or green team<br>➢ Presence of a designated team leader.<br>➢ Team support for environmental projects and activities.<br>➢ Engagement in the organizations sustainability development and management<br>➢ Teams engagement in strategy and policy making of the firm.<br>➢ Teams engagement in the risk management of the firm. |
| Corporate Sustainability Practices | ➢ Risk management strategy<br>➢ Investment in new technology and infrastructure update<br>➢ Complying with CSR<br>➢ Volunteer work program/donations for charity.<br>➢ Trainee program/career development policy<br>➢ Health and safety at work<br>➢ Response and participation in societal and ethical demands<br>➢ Investment in Security information/IT solution and HR<br>➢ Focus on local suppliers and supply chain<br>➢ Possibilities of generation of jobs<br>➢ Activities in strategic new market |

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
