# Peer review of "The Impact of Chief Sustainability Officers on Environmental Performance of Korean Listed Companies: The Mediating Role of Corporate Sustainability Practices"

_sustainability, doi:10.3390/su152014819_

Round 1

Reviewer 1 Report

Thank you for an interesting topic on how CSO impacts environmental performance of organisations via the mediation of CSP. 

Most importantly, I have my doubts about mediation in such a context. Is it scientifically sound to test mediation of CSP on environmental performance. All of the literature cited in the article indicates positive correlation between CSO, CSP and environmental performance and that CSO is important when implementing CSP and improving environmental performance. Or maybe the theoretical grounding is not so strong to make it clear why such hypothesis (H4) was raised. A good example is provided for testing mediation that has a strong logical grounding (Rahman et. al, 2021) that is clear from the title. However, the present article lacks such clear distinction in the logical grounding.

Additional points for consideration:

1. Moreover, how construct items were selected for each variable (Appendix 1)? Did you measure the environmental performance or just looked into whether they provided statistics over the years?

2. Formatting is required for tables, formulas and graphs.

3. The results of regression analysis should be provided in the form of graph as it is usualy easier for the reader to understand. 

4. I missed the information about the software used for calculations. 

5. Referencing must be reviewed. If the authors have not used any referencing software, I would highly suggest to start using any one of them.

The article is written in clear English and is easy to follow. However, English proofreading should be done. 

Reviewer 2 Report

This research investigates the potential relationship between the appointment of CSOs and the environmental performance of Korean listed firms, with corporate sustainability practices (CSP) serving as a mediator.

This topic is interesting because it examines the managerial and governance role of CSOs in driving environmental performance improvements and recommends their incorporation into decision-making processes to promote sustainability practices inside companies. However, the article needs be revised before it can be published.

The figure depicting the conceptual framework for this study adds little value to the study's framework.

Some additional work is necessary to test the model's robustness (the authors could have performed sensitivity analyses by using different regression models or alternative specifications to test the robustness of their findings), and some external validation in conclusions part (the authors could have compared their findings with previous studies or conducted cross-validation with data from different time periods or regions to validate the generalizability of their findings) should be provided.

Some passages (e.g., the first paragraph on page 13) contain grammatical errors, so the authors should carefully revise the English.

Reviewer 3 Report

1) Though the authors have tried to discuss the research gap, however, it is also requested to add how the current research model is different or unique in comparison to previous studies for better understanding of the research gaps.

2) A well-written literature review. The authors are requested to add some literature from 2023. Presently, there is no literature from 2023. 

3) In measurement, the authors have mentioned several times that all the items were selected using content analysis. Please discuss in detail the item selection criteria via content analysis for clarity, i.e., from how many items you have chosen the mentioned items and how, please justify.

4) As mentioned the control variables are chosen based on previous research thus please provide some previous research examples or justify the reasons for selecting these control variables. 

5) Please add theoretical contributions as the present study has used two theories i.e., Upper echelons theory and  Legitimacy Theory.

Best wishes. 

Round 2

Reviewer 1 Report

Thank you for explaining the hypothesis more thoroughly. 

However, the introduction now is too long. The model is still only presented in the primary form and not included in the results.

The formatting of tables, formulas, references was not improved.

English proofreading must be conducted.

Author Response

We appreciate your valuable comments, which we believe improved the quality of our paper.

Regarding your comment about our model not being presented in our research results, we would like to clarify that our models have been incorporated into the explanation of our research results in the regression result section and also included in regression result table 3.

We have made both major and minor corrections, including thorough proofreading to rectify grammatical errors.  We have also rephrased numerous sentences to enhance the clarity of our research paper. Furthermore, we have reformatted our result tables and utilized Microsoft equation tools to present our regression equation models. Additionally, we have improved our referencing to align with the accepted format.

Kindly be aware that the most recent corrections are indicated in blue text and highlighted in yellow, whereas, the earlier revisions are in red text.

Reviewer 2 Report

the article can be published, but some minor editing corrections are still required (and the authors should use Microsoft Equation tool for the equations).

the article can be published, but some minor editing corrections are still required (and the authors should use Microsoft Equation tool for the equations).

Author Response

We appreciate your valuable comments, which we believe improved the quality of our paper.

We have made both major and minor corrections, including thorough proofreading to rectify grammatical errors.  We have also rephrased numerous sentences to enhance the clarity of our research paper. Furthermore, we have reformatted our result tables and utilized Microsoft equation tools to present our regression equation models. Additionally, we have improved our referencing to align with the accepted format.

Kindly be aware that the most recent corrections are indicated in blue text and highlighted in yellow, whereas, the earlier revisions are in red text.